# Could Mid- to Late-Onset Glaucoma Be Associated with an Increased Risk of Incident Dementia? A Nationwide Retrospective Cohort Study

**DOI:** 10.3390/jpm13020214

**Published:** 2023-01-26

**Authors:** Dong-Kyu Kim, So Yeon Lee

**Affiliations:** 1Institute of New Frontier Research, Division of Big Data and Artificial Intelligence, Chuncheon Sacred Heart Hospital, Hallym University College of Medicine, Chuncheon 24252, Republic of Korea; 2Department of Otorhinolaryngology-Head and Neck Surgery, Chuncheon Sacred Heart Hospital, Hallym University College of Medicine, Chuncheon 24252, Republic of Korea; 3Department of Ophthalmology, Nune Eye Hospital, Seoul 06198, Republic of Korea

**Keywords:** glaucoma, dementia, Alzheimer’s, Parkinson’s, risk

## Abstract

This study aimed to investigate the possible link between glaucoma and dementia using a nationwide cohort sample of data. The glaucoma group (*n* = 875) included patients diagnosed between 2003 and 2005, aged over 55 years; the comparison group was selected using propensity score matching (*n* = 3500). The incidence of all-cause dementia was 18.67 (7014.7 person-years) among those with glaucoma aged over 55 years. The glaucoma group developed all-cause dementia more frequently than those in the comparison group (adjusted hazard ratio (HR) = 1.43, 95% confidence interval (CI), 1.17–1.74). In a subgroup analysis, primary open-angle glaucoma (POAG) showed a significantly increased adjusted HR for all-cause dementia events (1.52, 95% CI: 1.23–1.89), whereas we could not find any significant association in patients with primary angle-closure glaucoma (PACG). Additionally, POAG patients showed an increased risk of the development of Alzheimer’s disease (adjusted HR = 1.57, 95% CI, 1.21–2.04) and Parkinson’s disease (adjusted HR = 2.29, 95% CI, 1.46–3.61), but there was no significant difference in PACG patients. Moreover, the risk of Alzheimer’s disease and Parkinson’s disease was higher within 2 years of POAG diagnosis. Although our findings have some limitations, such as confounding factor bias, we suggest that clinicians should pay attention to the early detection of dementia in patients with POAG.

## 1. Introduction

Glaucoma is characterized by optic neuropathy with progressive degeneration of retinal ganglion cells, which represents the visual field defect [1]. Its typical symptoms are a gradual loss of peripheral vision that is followed by progressive loss of central vision. Although early detection and management of glaucoma can alleviate the disease status and economic burdens of glaucoma greatly, early-stage glaucoma generally is asymptomatic; thus, it is a major leading cause of blindness in the developed world [2]. It mostly affects adults over 40, but young adults, children, and even infants can have it. Currently, the two most common forms of glaucoma are primary open-angle glaucoma (POAG) and primary angle-closure glaucoma (PACG). The goal of glaucoma treatment is to maintain the patient’s visual function and related quality of life at a sustainable cost. Thus, the cost of treatment in terms of inconvenience and side effects, as well as financial implications for the individual and society, requires careful evaluation. On these days, most patients with early to moderate glaucoma damage could have suitable visual function and a modest reduction in quality of life. Dementia is a neurodegenerative disease group in which there is deterioration in cognitive function beyond what might be expected from the usual consequences of biological aging. Among those, Alzheimer’s disease and Parkinson’s disease are the most common type of dementia [3], whereas these usually show a long asymptomatic period. Thus, at the time of diagnosis, affected patients often suffered from extensive and irreversible damage [4]. Like other progressive brain diseases, they are associated with a buildup of certain proteins in the brain. Generally, Alzheimer’s disease always causes dementia, whereas Parkinson’s disease, a movement disorder, can sometimes cause dementia. For these reasons, both glaucoma and dementia have common pathologic features in terms of neurodegenerative conditions characterized by neuronal loss leading to cognitive and visual dysfunction, respectively. Additionally, both diseases become more prevalent according to increased age. To date, due to certain pathogenic and age-prevalence similarities, several epidemiologic studies have shown the potential risk of developing dementia in patients with glaucoma [5,6,7]. One Taiwan cohort study reported that POAG is a significant predictor for the development of Alzheimer’s disease, but POAG is not a predictor of Parkinson’s disease [5]. Other Taiwan cohort studies also described that female dementia patients showed a higher proportion of prior POAG than controls [6]. Meanwhile, one cohort study from South Korea revealed that POAG was associated with an increased risk of developing Alzheimer’s disease; however, there was no positive association between POAG and Parkinson’s disease [7]. However, these studies have some critical limitations regarding study design, including the presence of wash-our period, diagnosis age of glaucoma, and the date of enrolment. Therefore, to further investigate the relationship between the two diseases, we examined the association of mid- to late-onset glaucoma with the prospective risk of dementia using a representative sample from the National Sample Cohort data in the Republic of Korea.

## 2. Materials and Methods

This study was also approved by the Institutional Review Board of Hallym Medical University Chuncheon Sacred Hospital (IRB No. 2022-09-001), and the need for written informed consent was waived because the Korean National Health Insurance Service cohort (KNHIS) data set consisted of deidentified secondary data for research purposes. In terms of data availability, the authors confirm that the data supporting the findings of this study are available within the article.

### 2.1. Database

South Korea has had a single-payer national health system covering the entire Republic of the Korean population since 1989. An insured individual pays for national health insurance, which is proportional to the individual’s income, and each Republic of Korea is assigned a unique identification number at birth. With the integration of medical aid data into the KNHIS database in 2006, this database comprises the entire population of the Republic of Korea. For these reasons, the claims data in the KNHIS cannot be omitted or duplicated. Therefore, usage of the KNHIS database eliminates selection bias. Additionally, all disease diagnostic codes were identified using the Korean Classification of Disease, Fifth Edition modification of the International Classification of Disease and Related Health Problems, 10th revision (ICD-10). Thus, in this study, we utilized a database of the representative cohort sample consisting of 1,025,340 adults obtained from the KNHIS healthcare claims data. In 2002, we conducted stratified random sampling was performed among the Republic of Korea population of 46 million with 1476 strata by age (18 groups), sex (two groups), and income levels (41 groups: 40 health insurance and one medical aid beneficiary). Thus, we investigated the present study using the KNHIS database collected from 2002 to 2013, comprising information from a nationally representative sample of 1,025,340 random individuals, accounting for approximately 2.2% of the Republic of the Korean population in 2002.

### 2.2. Study Design

Briefly, Figure 1 shows the study design and the process of enrolment for the study participants. We had a washout period of one year (January to December 2002) to exclude those with a risk of developing dementia before glaucoma diagnosis. First, the glaucoma group was defined as the presence of diagnostic code (H40.1 and H40.2) more than two times within the index period and over age 55 years. These cases of POAG were all diagnosed by certified ophthalmologists. We also excluded patients (1) aged < 55 years, (2) who died during the index period, and (3) diagnosed with dementia before glaucoma diagnosis. Next, we identified non-glaucoma participants as a comparison group (non-glaucoma) using propensity score-matching methodology from the remaining cohort registered in the database as four participants without cancer for each cancer patient. Additionally, we selected each participant in the comparison group who matched with each patient in the glaucoma group in terms of all independent variables and the date of enrolment (glaucoma diagnosis). The primary endpoints in this study were defined as the specific event (dementia: Alzheimer’s [F00, G30], Parkinson’s disease [G20], other types of dementia [F01, F02, F03]) until 31 December 2013. If patients had no events until the final following period of this database, we censored this time point. In this study, to elaborate the analysis, we set some patients’ detailed characteristics as dependent variables (age, sex, residence, and household income). Additionally, we obtained information on the comorbidities of each individual and categorized the comorbidities using the Charlson comorbidity index (CCI), which is a weighted index of categorizing comorbidities of patients. The CCI is a weighted index to predict the risk of death within 1 year of hospitalization for patients with specific comorbid conditions. Nineteen conditions were included in the index. All independent variables are classified as follows: age (55–64, ≥65 years), sex (male, female), residence (Seoul, the largest metropolitan region in South Korea; 2nd area: other metropolitan cities in the Republic of Korea; and 3rd area: small cities and rural areas), household income (low: ≤30%, middle: 30.1–69.9%, and high: ≥70% of the median), and three comorbidity status (CCI: 0, 1, ≥2).

### 2.3. Statistical Analysis

We assessed the incidence rate as a measure of the frequency with which a specific disease or other incident events appears over a certain period. The overall incidence was expressed as per 1000 person-years, which is the following three cases: First, if the participant died, the number of years from the initial specific events diagnosis to the date of death; second, if specific events appeared, the number of years from the initial glaucoma diagnosis to the date of the first diagnosis of specific events; finally, if there are no events, the number of years from the date of initial cancer diagnosis to the final following period. Additionally, we used Cox proportional hazard regression analyses to calculate the hazard ratio (HR) and 95% confidence intervals (CI), adjusted for the other independent variables. During the follow-up period, the Kaplan–Meier method was used to calculate the specific disease-free survival rates among groups. All statistical analyses were performed using R version 4.0.5 (URL https://www.R-project.org/ accessed on 1 March 2021). *p*-values of <0.05 were considered statistically significant.

## 3. Results

The present study consisted of 875 participants with mid- to late-onset glaucoma and 3500 participants without glaucoma. We followed these participants in both groups for 10 years. We also confirmed that the distributions of all independent variables were similar between the groups using the balance plot technique. It indicated that the matching between the two groups was appropriate. The detailed characteristics of the study participants in each group are shown in Table 1.

In this study, 7014.7 person-years in the glaucoma group and 2893.5 person-years in the comparison group were evaluated for dementia events. The overall incidence of dementia was evaluated at 18.67 per 1000 person-years in the glaucoma group (Table 2). Additionally, for subgroup analysis, we assessed Alzheimer’s disease, Parkinson’s disease, and other types of dementia. We detected that, in the glaucoma group, the overall incidence was 11.85 in Alzheimer’s disease, 4.25 in Parkinson’s disease, and 7.88 in other types of dementia, respectively (Table 2). All subgroups showed a higher incidence rate of each subtype of dementia in mid- to late-onset glaucoma patients than in comparison.

We analyzed the risk of the subsequent development of dementia using univariate and multivariate Cox regression models (Table 3). After adjusting for all independent variables, we found that glaucoma was significantly associated with the development of dementia (adjusted HR = 1.43, 95% CI, 1.17–1.74). The risk of incident dementia events is also significantly increased in POAG patients but not in PACG patients.

In Figure 3, we detected that patients with POAG showed a significantly increased risk for dementia development, but there was no association in PACG patients. Additionally, when we investigated the risk of dementia according to subtype, we found that POAG patients were significantly associated with the development of Alzheimer’s disease (adjusted HR = 1.57, 95% CI, 1.21–2.04) and Parkinson’s disease (adjusted HR = 2.29, 95% CI, 1.46–3.61); however, there was no significant difference in the risk of each subtype of dementia in PACG patients (Figure 2). Meanwhile, other types of dementia showed no association in both POAG or PACG patients.

The Kaplan–Meier survival analysis revealed that patients in the glaucoma group, specifically in POAG, presented a more frequent incidence of all-cause dementia, Alzheimer’s disease, and Parkinson’s disease events than those in the control group (Figure 3).

Moreover, in the analysis of HRs over time, the risk of Alzheimer’s disease and Parkinson’s disease development in patients with POAG was higher within two years after glaucoma diagnosis (Table 4). Meanwhile, two years after glaucoma diagnosis, the risk level of Alzheimer’s disease and Parkinson’s disease development in POAG patients showed similar during the follow-up period.

Furthermore, we performed the subgroup analysis of HRs according to sex and comorbidities. We detected that the risk of Alzheimer’s disease was higher in female patients with POAG, whereas male patients showed a higher risk of Parkinson’s disease (Table 5). In terms of comorbidities, the adjusted HR of Alzheimer’s disease and Parkinson’s disease development in POAG patients was significantly increased in the comorbidities group compared to the non-comorbidities group (Table 6).

## 4. Discussion

Evidence obtained from many previous studies may show a possible link between two diseases based on potential pathophysiological mechanisms. To our knowledge, the epidemiologic relationship between neurodegenerative diseases and glaucoma has been widely investigated. Although some studies have not found an increased risk of dementia in patients with glaucoma [8,9], many studies have demonstrated that POAG has a potentially increased risk of Alzheimer’s disease [5,6,7]. However, some limitations of these epidemiologic studies were that there was no washout period to eliminate the preexisting dementias, glaucoma patients include a relatively young age, and the matching control group is not clear. In this study, using longitudinal data from a nationwide cohort, we demonstrated an increased risk of not only Alzheimer’s disease but also Parkinson’s disease development in POAG patients aged over 55 years during a 10-year follow-up period compared with a comparison group (non-glaucoma). Interestingly, we also found that patients with POAG showed a higher risk of Alzheimer’s disease and Parkinson’s disease development within two years after glaucoma diagnosis. Two years later, on glaucoma diagnosis, the risk level of Alzheimer’s disease and Parkinson’s disease development in POAG patients slightly decreased and then continued to a constant level during the follow-up period.

It is known that various evidence exists possible linking between the two diseases, including structural abnormalities, specifically degenerative changes within ganglion cells. Besides Similar degenerative changes, neurodegenerative diseases and POAG have possible overlapping pathophysiologic mechanisms. Although the exact pathogenesis is still unclear, elevated intraocular pressure (IOP) is a major risk factor for the development and progression of POAG [10]. The difference in IOP and CSF pressure across the lamina cribrosa, the translaminar pressure difference, is an important factor in causing more optic nerve damage [11,12]. Additionally, some neurotoxic substances such as β-amyloid and tau protein were observed in both neurodegenerative disease and glaucoma [13,14,15]. Another possible explanation for the possible link between the two diseases is that vascular factors, such as hypertension, DM, and hyperlipidemia, are one of the major risk factors for developing each disease [16,17,18]. Similar to previous reports, [5,6,7] our findings showed the association between POAG and Alzheimer’s disease. However, contrary to these reports, we also find the increased subsequent development of Parkinson’s disease in POAG.

Parkinson’s disease is most commonly known for affecting function and movement, though it also affects cognition, particularly as the disease progresses. It is a neurodegenerative disorder of the brain, such as Alzheimer’s disease, and also has similar neurotoxic substances and vascular risk factors [19,20,21]. However, prior studies showed no association between POAG and Parkinson’s disease during the follow-up period [5,22]. However, one study included a young age group in POAG patients, and the other study only included patients who had received anti-glaucoma medication or undergone glaucoma surgery during the study period. We thought that these issues might play a confounding variable in evaluating the risk of incident Parkinson’s disease events. Thus, in the present study, we identified the risk of dementia in all glaucoma patients over 55 years who were diagnosed by certified ophthalmologists. Consistent with our findings, some studies demonstrated that Parkinson’s disease was more likely to develop glaucoma-like VF defects and observed reduced RNFL thickness [23,24,25,26].

PACG is a different kind of glaucoma compared with POAG. It is known that the primary pathological cause of PACG is angle closure, and high IOP is secondarily induced due to angle closure. Thus, PACG patients commonly detected a shallow anterior chamber, thickened lens, and hyperopic refractive error. Meanwhile, the cause of POAG is mainly obstruction of the aqueous humor pathway due to trabecular meshwork degeneration. This angle closure results in the prevention of aqueous humor exit and is followed by IOP, which is thought to damage the optic nerve. PACG shows its distinctive anatomical characteristic, such as a narrow anterior chamber angle, which induces its unique pathological process [27,28]. Meanwhile, several previous cohort studies showed more association of dementia in POAG, not in PACG [5,6,7]. Our findings were also consistent with previous cohort studies.

Interestingly, we found that the risk of Alzheimer’s disease and Parkinson’s disease development was relatively higher within two years after the diagnosis of POAG. Although we could not know why the risk is higher in the early period after glaucoma diagnosis, it means that early detection of Alzheimer’s disease and Parkinson’s disease is clinically important to patients with POAG aged over 55 years. We also found that the risk of Alzheimer’s disease and Parkinson’s disease was different according to sex or comorbidities. Generally, female is nearly twice as likely as male to develop Alzheimer’s disease, and the risk of developing Parkinson’s disease is twice as high in males than in females. Meanwhile, it is known that comorbidity in neurodevelopmental disorders is pronounced. However, in this study, we could not determine whether these findings are a clue to a possible link between the two diseases or only a temporal incidental finding. Thus, we need further studies to prove these findings.

This study has several limitations. First, this study could not present the direct mechanism between dementia and POAG due to our study design, in which the baseline characteristics of the individuals are limited to a previous database. Thus, we could not confirm whether our findings are a causal relationship or temporal incidence. Second, the diagnosis of glaucoma and dementia was based on the ICD-10 diagnostic code, not medical records that include details such as the patient’s medical history and the results of neurocognitive questionnaires. It means that we could not determine the severity of these diseases and also have a misclassification bias. At present, to overcome this issue, we only included glaucoma or dementia patients who were diagnosed by ophthalmologists or neurologists. Third, we did not consider whether patients received anti-glaucoma medication or underwent glaucoma surgery during the study period; thus, these variables may influence our findings. Fourth, in this study, the exact onset time of each disease is unclear, although we considered it as the age at the first hospital visit for each disease. Finally, in this study, we adjusted several variables, which are commonly known as these variables, that could influence the primary outcome events. Thus, the dependent variables in this study are age, sex, residence, and household income. However, other dependent variables could not adjust in this study, and then, among those, some may become confounding variables. For example, we could not adjust personal health data, including the body mass index, smoking history, and alcohol consumption, because we could not access this information. Thus, these might be played as confounding factors, and our findings inevitably have their own limitations.

Nevertheless, our study also has several unique advantages. First, our database could provide us with an effective analyze all events associated with glaucoma and dementia because this cohort had a long follow-up period and represented the Republic of Korea population. Second, the reliability of the KNHIS database has been validated, which showed a similar prevalence of 20 major diseases for each of the years assessed; thus, we are able to ascertain the reliability of the KNHIS data as “fair to good” [29,30,31]. Third, among databases, we just selected patients who were first diagnosed with neurodegenerative diseases by neurologists and glaucoma by ophthalmologists. We thought it enhanced the accuracy of defining the study group. Finally, this study minimized the surveillance bias on the risk of dementia in POAG patients because we selected sociodermographically matched controls in the cohort database.

## 5. Conclusions

In the present study, we identified the association between mid- to late-onset glaucoma and the risk of dementia after adjusting for clinical and demographic factors. Our findings suggest an increased risk of all-cause dementia, Alzheimer’s disease, and Parkinson’s disease events in patients with POAG aged over 55 years; however, no significant association was observed in patients with PACG aged over 55 years. Additionally, the risk of Alzheimer’s disease and Parkinson’s disease is relatively higher within two years of POAG diagnosis. Therefore, given the potential link between the two diseases, clinicians should be aware of the potential development of dementia in patients with POAG and recommend neurologic consulting to ensure early detection of neurodegenerative diseases during the prodromal period.

## Figures and Tables

**Figure 1 jpm-13-00214-f001:**
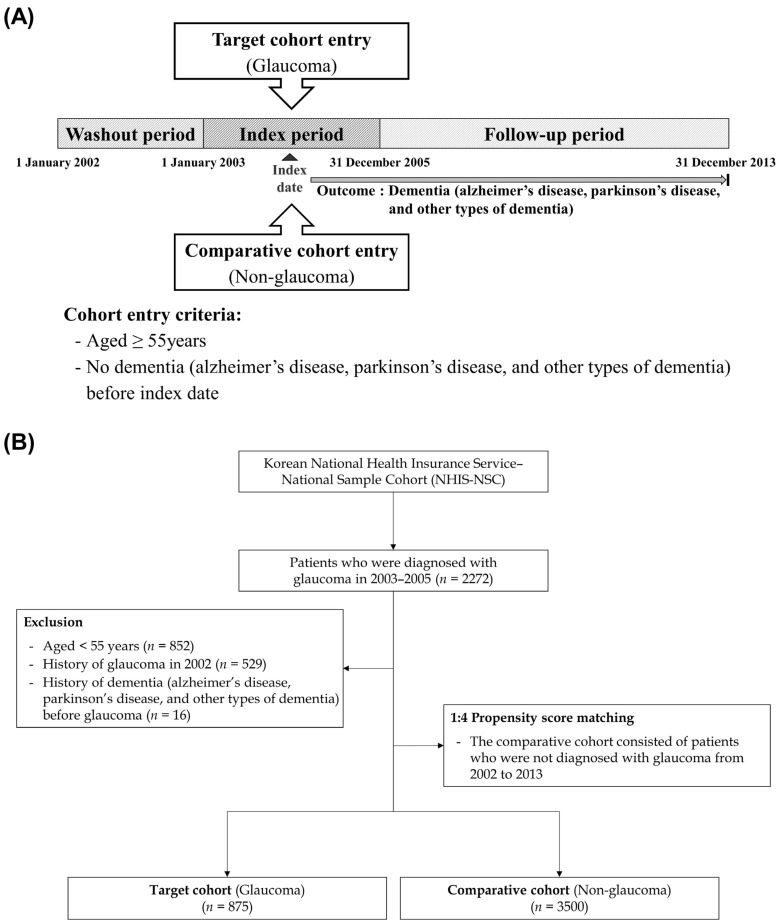
(**A**) Schematic study design; (**B**) process of study enrolment.

**Figure 2 jpm-13-00214-f002:**
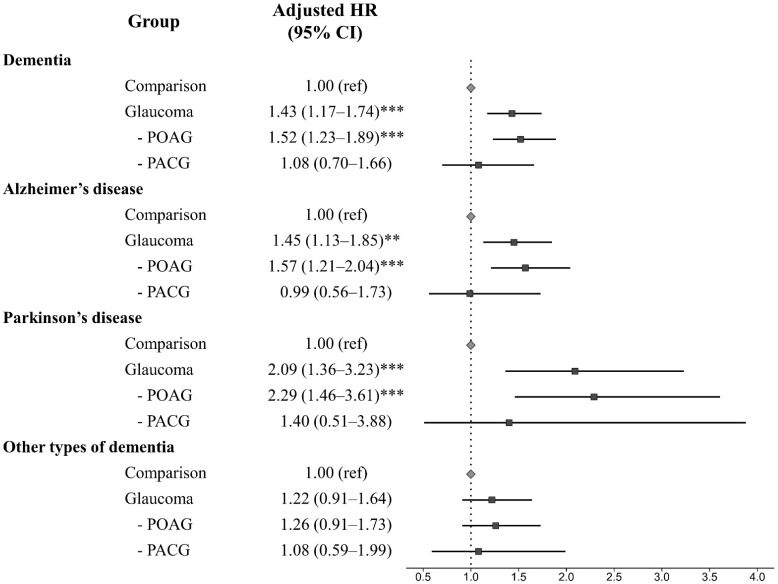
The adjusted hazard ratio plot of each dementia subtype among groups. POAG, primary open-angle glaucoma; PACG, primary angle-closure glaucoma; HR, hazard ratio; CI, confidence interval. ** *p* < 0.010 and *** *p* < 0.001.

**Figure 3 jpm-13-00214-f003:**
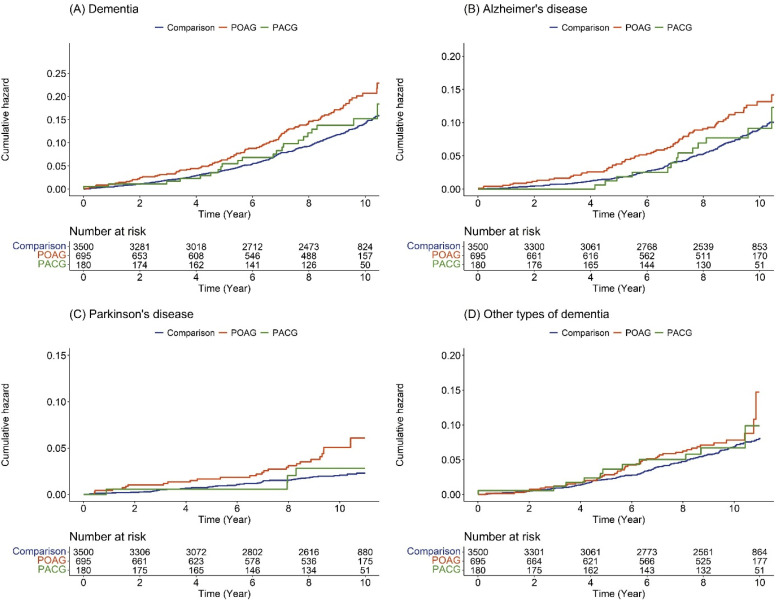
Cumulative hazard plot of specific diseases among comparison, POAG, and PACG: (**A**) dementia; (**B**) Alzheimer’s disease; (**C**) Parkinson’s disease; (**D**) other types of dementia. POAG, primary open-angle glaucoma; PACG, primary angle-closure glaucoma.

**Table 1 jpm-13-00214-t001:** Characteristics of the study participants in each group.

Variables	Comparison (*n* = 3500)	Glaucoma (*n* = 875)	*p*-Value
Sex			1.000
Male	1572 (44.9%)	393 (44.9%)	
Female	1928 (55.1%)	482 (55.1%)	
Ages (years)			1.000
55–64	1424 (40.7%)	356 (40.7%)	
≥65	2076 (59.3%)	519 (59.3%)	
Residence			1.000
Seoul	724 (20.7%)	181 (20.7%)	
Second area	820 (23.4%)	205 (23.4%)	
Third area	1956 (55.9%)	489 (55.9%)	
Household income			1.000
Low (0–30%)	704 (20.1%)	176 (20.1%)	
Middle (30–70%)	1172 (33.5%)	293 (33.5%)	
High (70–100%)	1624 (46.4%)	406 (46.4%)	
CCI			1.000
0	1660 (47.4%)	415 (47.4%)	
1	900 (25.7%)	225 (25.7%)	
≥2	940 (26.9%)	235 (26.9%)	

**Table 2 jpm-13-00214-t002:** The incidence rate of dementia between glaucoma and comparison (non-glaucoma) groups.

Variables	N	Case	Person Year	Incidence
Dementia				
Comparison	3500	393	28,935.3	13.58
Glaucoma	875	131	7014.7	18.67
Alzheimer’s disease				
Comparison	3500	251	29,411.2	8.53
Glaucoma	875	85	7175.5	11.85
Parkinson’s disease				
Comparison	3500	61	29,776.4	2.05
Glaucoma	875	31	7294.7	4.25
Other types of dementia				
Comparison	3500	194	29,504.8	6.58
Glaucoma	875	57	7231.1	7.88

**Table 3 jpm-13-00214-t003:** The risk of incident dementia events between mid- to late-life glaucoma and comparison (non-glaucoma) groups.

Variables	N	Case	Unadjusted HR (95% CI)	Adjusted HR (95% CI)
Comparison	3500	393	1.00 (ref)	1.00 (ref)
Glaucoma	875	131	1.44 (1.18–1.75) ***	1.43 (1.17–1.74) ***
POAG	695	109	1.52 (1.23–1.88) ***	1.52 (1.23–1.89) ***
PACG	180	22	1.13 (0.74–1.74)	1.08 (0.70–1.66)

POAG, primary open-angle glaucoma; PACG, primary angle-closure glaucoma; HR, hazard ratio; CI, confidence interval. *** *p* < 0.001.

**Table 4 jpm-13-00214-t004:** Risk of dementia subtype in primary open-angle glaucoma (POAG) patients by time.

Time (Year)	Dementia		Alzheimer’s Disease		Parkinson’s Disease		Other Types of Dementia	
	No. of Event	Adjusted HR (95% CI)	No. of Event	Adjusted HR (95% CI)	No. of Event	Adjusted HR (95% CI)	No. of Event	Adjusted HR (95% CI)
1	7	2.03 (0.84–4.90)	4	4.07 (1.09–15.19) *	3	2.88 (0.69–12.07)	1	0.64 (0.08–5.11)
2	16	2.16 (1.20–3.89) *	8	2.69 (1.14–6.36) *	7	4.29 (1.55–11.84) **	4	1.26 (0.42–3.76)
3	22	1.69 (1.04–2.75) *	11	2.25 (1.10–4.57) *	7	1.99 (0.83–4.81)	8	1.36 (0.62–2.99)
4	29	1.55 (1.02–2.36) *	17	2.20 (1.25–3.90) **	9	2.01 (0.92–4.37)	11	1.22 (0.63–2.37)
5	40	1.58 (1.11–2.25) *	24	2.19 (1.36–3.55) **	11	2.03 (1.00–4.09) *	18	1.31 (0.78–2.20)
6	54	1.63 (1.20–2.21) **	33	2.05 (1.37–3.08) ***	12	1.73 (0.90–3.35)	26	1.52 (0.98–2.36)
7	68	1.50 (1.14–1.97) **	41	1.70 (1.19–2.42) **	15	1.74 (0.96–3.14)	32	1.36 (0.92–2.01)
8	84	1.59 (1.24–2.03) ***	53	1.74 (1.27–2.38) ***	19	2.00 (1.17–3.42) *	37	1.33 (0.92–1.91)
9	94	1.48 (1.18–1.86) ***	62	1.60 (1.20–2.12) **	22	1.95 (1.19–3.20) **	41	1.25 (0.89–1.76)
10	103	1.49 (1.19–1.85) ***	67	1.52 (1.16–2.00) **	26	2.25 (1.42–3.57) ***	43	1.20 (0.86–1.68)
11	109	1.52 (1.23–1.89) ***	72	1.57 (1.21–2.04) ***	27	2.29 (1.46–3.61) ***	46	1.26 (0.91–1.73)

HR, hazard ratio; CI, confidence interval. * *p* < 0.05, ** *p* < 0.010, and *** *p* < 0.001.

**Table 5 jpm-13-00214-t005:** Risk of Alzheimer’s disease and Parkinson’s disease in POAG patients according to the sex.

Sex	Male		Female	
	Comparison	POAG	Comparison	POAG
Alzheimer’s disease				
Unadjusted HR (95% CI)	1.00 (ref)	1.31 (0.92–1.85)	1.00 (ref)	2.18 (1.45–3.27) ***
Adjusted HR (95% CI)	1.00 (ref)	1.27 (0.90–1.81)	1.00 (ref)	2.16 (1.43–3.24) ***
Parkinson’s disease				
Unadjusted HR (95% CI)	1.00 (ref)	3.01 (1.70–5.36) ***	1.00 (ref)	1.57 (0.74–3.33)
Adjusted HR (95% CI)	1.00 (ref)	2.94 (1.65–5.23) ***	1.00 (ref)	1.57 (0.74–3.33)

*** *p* < 0.001.

**Table 6 jpm-13-00214-t006:** Risk of Alzheimer’s disease and Parkinson’s disease in POAG patients according to the comorbidities.

CCI	0		1		≥2	
	Comparison	POAG	Comparison	POAG	Comparison	POAG
Alzheimer’s disease						
Unadjusted HR (95% CI)	1.00 (ref)	1.51 (0.88–2.60)	1.00 (ref)	1.87 (1.12–3.10) *	1.00 (ref)	1.49 (1.03–2.17) *
Adjusted HR (95% CI)	1.00 (ref)	1.52 (0.88–2.61)	1.00 (ref)	1.84 (1.11–3.05) *	1.00 (ref)	1.47 (1.01–2.14) *
Parkinson’s disease						
Unadjusted HR (95% CI)	1.00 (ref)	1.43 (0.57–3.57)	1.00 (ref)	1.78 (0.70–4.48)	1.00 (ref)	3.52 (1.84–6.77) ***
Adjusted HR (95% CI)	1.00 (ref)	1.41 (0.57–3.53)	1.00 (ref)	1.75 (0.69–4.43)	1.00 (ref)	3.48 (1.81–6.69) ***

POAG, primary open-angle glaucoma; CCI, Charlson comorbidity index; HR, hazard ratio; CI, confidence interval. * *p* < 0.05 and *** *p* < 0.001.

## Data Availability

The authors confirm that the data supporting the findings of this study are available within the article.

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
