# Peer review of "Could Mid- to Late-Onset Glaucoma Be Associated with an Increased Risk of Incident Dementia? A Nationwide Retrospective Cohort Study"

_jpm, 2023, doi:10.3390/jpm13020214_

Round 1

Reviewer 1 Report

1. Please describe the type of manuscript in the title.

2. At the first presentation, every abbreviation should be fully described.

3. Confounding variables.

a. How were confounding variables assessed?

b. The authors should describe the significant confounding variables in their study. A short description in the abstract and a full explanation in the discussion should be given.

c. A direct association between the pathologies can not be written because the baseline characteristics of the individuals are limited to a previous database.

4. The reviewer would recommend grayscale for Figures 1 & 2.

Author Response

1. Please describe the type of manuscript in the title.

Answer: As you comment, we modified the title as follows: “Could mid- to late-onset glaucoma associate with an increased risk of incident dementia? A Nationwide Cohort Study”

2. At the first presentation, every abbreviation should be fully described.

Answer: As you comment, we added the full term before using the abbreviation.

3. Confounding variables.

a. How were confounding variables assessed?

Answer: In this study, we adjusted several variables, which are commonly known as these variables that could influence the primary outcome events. Thus, the dependent variables in this study are age, sex, residence, and household income. However, other dependent variables could not adjust in this study and then, among those, some may become confounding variables.

b. The authors should describe the significant confounding variables in their study. A short description in the abstract and a full explanation in the discussion should be given.

Answer: We added more description in the abstract and the section of the discussion.

c. A direct association between the pathologies can not be written because the baseline characteristics of the individuals are limited to a previous database.

Answer: We totally agreed with your opinion and added more description in the section of discussion as a limitation.

4. The reviewer would recommend grayscale for Figures 1 & 2.

Answer: We modified the figures as you recommend.

Reviewer 2 Report

Dear Editor,

Firstly, I want to thank you for giving me this opportunity. I have carefully read and reviewed this “Article” type of paper: “Could mid to late-life glaucoma associate with an increased 2 risk of incident dementia?”

The article is written about to investigate the relationship and the association of mid to late-life glaucoma with the prospective risk of dementia, using a representative sample from the National Sample Cohort data in South Korea.

I quite liked Figure 2 and found it useful. Figure 3 is really useful! The numbers 3500 and 875 seem quite sufficient for study.

The research content is in accordance with the aim and scope of the journal. There are problems in the text language such as intelligibility and grammatical errors.

The subject and methods are interesting, although there are a few issues that the authors need to clarify and correct. If the authors consider the suggestions I have mentioned below, the manuscript will be significantly improved.

Good luck and success.

1)     First of all, "This study is aimed to investigate the possible link between glaucoma and OSA using a nationwide cohort sample of data." First sentence in abstract section. What is the meaning of OSA??? If it means "Obstructive Sleep Apnea", what is the relevance?

2)     In Table 1, you have divided the age into two as 55-64 years ≥65 years. If possible, it may be more effective to create groups of 55-64, 65-85 and >85 and make comparisons here.

3)     Can you specify the types and rates of dementia? Vascular? Fronto Temporal Dementia? Lewy body dementia? Or is it not possible to obtain vascular, fronto temporal and lewy body dementias because it is a study created only with a retrospective diagnosis algorithm?

4)     Wouldn't it be better to have the title like this? : "Could mid- to late-onset glaucoma be associated with an increased risk of dementia?"

5)     In the limitations section, it should be noted that this is a retrospective design and retrospective study created with data obtained only through diagnoses.

6)     "Thus, with regard to the pathologic mechanisms, one prior study demonstrated an increased risk of sleep apnea in PACG, not POAG." Unnecessary information written for self-citation purposes. I strongly recommend take it out of the discussion.

7)     "Second, the reliability of the KNHIS database has been validated, which showed a similar prevalence of 20 major diseases for each of the years assessed; thus, we are able to ascertain the reliability of the KNHIS data as “fair to good”. Additionally, several retrospective cohort studies have already published a similar study design using this database." These sentences, created for self-citation purposes, pose serious ethical problems! I strongly recommend remove these. If you're going to talk about it, you can use simple sentences that aren't self-citation.

8)     Last but not least, "Third, we defined glaucoma and neurodegenerative diseases as diagnosed by ophthalmologists and neurologists, respectively." Yes, but unfortunately none of the neurologists you mentioned are involved in this study. Wouldn't that be disrespectful or inappropriate to neurologists?

Kind Regards

Author Response

1) First of all, "This study is aimed to investigate the possible link between glaucoma and OSA using a nationwide cohort sample of data." First sentence in abstract section. What is the meaning of OSA??? If it means "Obstructive Sleep Apnea", what is the relevance?

Answer: This is our mistake. We really appreciate your comment.

2) In Table 1, you have divided the age into two as 55-64 years ≥65 years. If possible, it may be more effective to create groups of 55-64, 65-85 and >85 and make comparisons here.

Answer: We agree with your opinion. However, the >85 age category has a very small number. So, it was very difficult to match between glaucoma and comparison groups. That’s why we used two age categories. Please, understand our setting for the age category.

3) Can you specify the types and rates of dementia? Vascular? Fronto Temporal Dementia? Lewy body dementia? Or is it not possible to obtain vascular, fronto temporal and lewy body dementias because it is a study created only with a retrospective diagnosis algorithm?

Answer: In this study, we classified into dementia group according to the ICD-10 diagnostic code. Specifically, Alzheimer's was defied as [F00, G30], Parkinson’s disease was defined as [G20], and other types of dementia were defined as [F01; vascular dementia, F02; Dementia in other diseases classified elsewhere, F03; Unspecified dementia]. Since the limitation for claims data study, we could not specify dementia type in more detail.

4) Wouldn't it be better to have the title like this? : "Could mid- to late-onset glaucoma be associated with an increased risk of dementia?"

Answer: We really appreciate your comment and modified it as you recommended.

5) In the limitations section, it should be noted that this is a retrospective design and retrospective study created with data obtained only through diagnoses.

Answer: As you comment, we emphasized more this limitation in the section of the discussion.

6) "Thus, with regard to the pathologic mechanisms, one prior study demonstrated an increased risk of sleep apnea in PACG, not POAG." Unnecessary information written for self-citation purposes. I strongly recommend take it out of the discussion.

Answer: As you comment, we removed this sentence.

7) "Second, the reliability of the KNHIS database has been validated, which showed a similar prevalence of 20 major diseases for each of the years assessed; thus, we are able to ascertain the reliability of the KNHIS data as “fair to good”. Additionally, several retrospective cohort studies have already published a similar study design using this database." These sentences, created for self-citation purposes, pose serious ethical problems! I strongly recommend remove these. If you're going to talk about it, you can use simple sentences that aren't self-citation.

Answer: As you comment, we removed this sentence. However, “the reliability of the KNHIS database has been validated~” is not a self-citation and very important sentence to explain our study's unique strength. Please, understand our choice.

8) Last but not least, "Third, we defined glaucoma and neurodegenerative diseases as diagnosed by ophthalmologists and neurologists, respectively." Yes, but unfortunately none of the neurologists you mentioned are involved in this study. Wouldn't that be disrespectful or inappropriate to neurologists?

Answer: I understand what you want to talk about. However, in this study, we just selected patients who were first diagnosed with dementia by neurologists. Also, we thought this point is very important because it supports that the diagnostic code was selected very accurately. Thus, we modified this sentence more clearly.

Round 2

Reviewer 2 Report

Dear Editor

Firstly, I want to thank you for giving me this opportunity again. I have carefully read and reviewed this REVISED “Article” type of paper: “Could mid- to late-onset glaucoma associate with an increased 2 risk of incident dementia? A Nationwide Cohort Study”

I have only one criticism right now! Specify Retrospective in the title! “A Nationwide Retrospective Cohort Study”

The authors really seem to have done their best. 

Now the manuscript has become more convenient and improved.

I do not see any inconvenience to endorse.

Kind regards

Good luck and success.

1)  Answer: This is our mistake. We really appreciate your comment.
OK

2) Answer: We agree with your opinion. However, the >85 age category has a very small number. So, it was very difficult to match between glaucoma and comparison groups. That’s why we used two age categories. Please, understand our setting for the age category.

OK

3) Answer: In this study, we classified into dementia group according to the ICD-10 diagnostic code. Specifically, Alzheimer's was defied as [F00, G30], Parkinson’s disease was defined as [G20], and other types of dementia were defined as [F01; vascular dementia, F02; Dementia in other diseases classified elsewhere, F03; Unspecified dementia]. Since the limitation for claims data study, we could not specify dementia type in more detail.

OK

4) Answer: We really appreciate your comment and modified it as you recommended.

OK

5) As you comment, we emphasized more this limitation in the section of the discussion.

OK

6) As you comment, we removed this sentence.

OK

7) Answer: As you comment, we removed this sentence. However, “the reliability of the KNHIS database has been validated~” is not a self-citation and very important sentence to explain our study's unique strength. Please, understand our choice.

OK

8) Answer: I understand what you want to talk about. However, in this study, we just selected patients who were first diagnosed with dementia by neurologists. Also, we thought this point is very important because it supports that the diagnostic code was selected very accurately. Thus, we modified this sentence more clearly.

OK

Kind Regards

Author Response

Thank you for your kind comment.

I revised the Title as you recommend.